# A Spatial Analysis of the Spread of Hyperendemic Sporotrichosis in the State of Rio de Janeiro, Brazil

**DOI:** 10.3390/jof8050434

**Published:** 2022-04-23

**Authors:** Eduardo Mastrangelo Marinho Falcão, Anselmo Rocha Romão, Mônica de Avelar Figueiredo Mafra Magalhães, José Berilo de Lima Filho, Antonio Carlos Francesconi do Valle, Francisco Inácio Bastos, Maria Clara Gutierrez-Galhardo, Dayvison Francis Saraiva Freitas

**Affiliations:** 1Instituto Oswaldo Cruz, Fundação Oswaldo Cruz, Rio de Janeiro 21040-90, RJ, Brazil; dayvison.freitas@ini.fiocruz.br; 2Instituto Nacional de Infectologia Evandro Chagas, Fundação Oswaldo Cruz, Rio de Janeiro 21040-90, RJ, Brazil; jose.berilo@ini.fiocruz.br (J.B.d.L.F.); antonio.valle@ini.fiocruz.br (A.C.F.d.V.); maria.clara@ini.fiocruz.br (M.C.G.-G.); 3Instituto de Comunicação e Informação Científica e Tecnologia em Saúde, Fundação Oswaldo Cruz, Rio de Janeiro 21040-90, RJ, Brazil; anselmo.romao@icict.fiocruz.br (A.R.R.); monica.magalhaes@icict.fiocruz.br (M.d.A.F.M.M.); francisco.inacio.bastos@hotmail.com (F.I.B.)

**Keywords:** sporotrichosis, zoonoses, *Sporothrix*, epidemiology, geographic information systems

## Abstract

Sporotrichosis is a subacute/chronic subcutaneous mycosis. Since the late 1990s, there has been a hyperendemic zoonotic transmission in the state of Rio de Janeiro, involving *Sporothrix brasiliensis*, the most virulent causative species, and a “belt” was described along the limits between the capital and its outskirts (“Baixada Fluminense”). This study analyzes the distribution of sporotrichosis using secondary data from the Notifiable Diseases Information System (Sinan) of the Rio de Janeiro State Health Department (SES/RJ) from 2011 to 2015 and from the INI Electronic Patient Record System (Sipec) from 2008 to 2015. Cases diagnosed since the onset of the hyperendemic exceed all previously reported case series of the disease and there is a progressive expansion in the state of Rio de Janeiro. The study suggests the spread of the mycosis to all regions of the state and the expansion of the previously described “belt”, despite public health measures and changes in its profile over the years, with great social impact.

## 1. Introduction

Sporotrichosis is a subacute/chronic subcutaneous mycosis, classically described as secondary to the traumatic inoculation of fungi of the genus *Sporothrix* [1,2], found abundantly as saprobes in soil and decaying vegetation throughout the world, in different climates and even in adverse conditions [3].

Several outbreaks of sporotrichosis originating from a common source have been described. The largest one occurred in South Africa, infecting approximately 3000 gold mine workers [4]. Hyperendemic regions have been recently described in China, Mexico, India, and Brazil. In China, 2113 cases in adults and 855 in children under 15 years of age have been reported, from 2010 to 2018, in Jilin province, mostly in males [5]. In India, the sub-Himalayan region is considered endemic, with approximately 300 cases over 18 years [6]. In Mexico, sporotrichosis is the most frequent and hyperendemic subcutaneous mycosis in the states of Jalisco and Michoacan [7,8].

In Brazil, hospitalizations and deaths have been documented all over the country [9]. However, sporotrichosis is not a nationally notifiable disease, except for some municipalities and states (Rio de Janeiro, Pernambuco, Paraíba, Minas Gerais and the Federal District).

Since the late 1990s, there has been a hyperendemic zoonotic transmission in the state of Rio de Janeiro (RJ) [2,10,11,12]. Most cases with molecular analysis of the causative agent are due to a more virulent species of the fungus, *Sporothrix brasiliensis* [13]. Most patients are middle-aged women involved in domestic exposure to cats while taking care of the house, but the hyperendemic highlighted vulnerable groups, especially people living with HIV, leading to hospitalizations and deaths [14,15].

In addition to RJ, in recent decades the presence of *S. brasiliensis* has been reported in other Brazilian states (Minas Gerais, São Paulo, Espírito Santo, Paraná, Rio Grande do Sul, Santa Catarina, Distrito Federal, Rio Grande do Norte, Pernambuco and Alagoas), with the possibility of more than one place of origin of the species being even suggested [16,17,18,19,20]. More recently, autochthonous cases of sporotrichosis by zoonotic transmission have also been reported in other South and Central American countries [21].

Several publications describe the clinical and epidemiological characteristics of zoonotic sporotrichosis in RJ, but the distribution of cases in the state was studied only until 2007. Silva et al. analyzed the spatial distribution of patients diagnosed at the Evandro Chagas National Institute of Infectious Diseases (INI), Oswaldo Cruz Foundation (FIOCRUZ), the main reference center in the state, describing for the first time a “sporotrichosis belt”. The belt was formed by high-density areas throughout the limits between the capital and the municipalities of Baixada Fluminense, located in the capital outskirts [22]. Since then, in 2011, a technical note implementing epidemiological surveillance in RJ was published [23] and, in 2013, a resolution for mandatory notification of human sporotrichosis in the state was approved [24].

The aim of this study was to analyze the distribution of sporotrichosis after the previously published study, using data from outpatients and inpatients of INI/FIOCRUZ, as well as including patients seen at other health units, to cover as accurately as possible the cases in the state of RJ. Although far from comprehensive, such an update is sorely needed since the diffusion process is far from being curbed, with a substantial burden for individuals and communities.

## 2. Materials and Methods

### 2.1. Study Design

This is a retrospective study using secondary data from the Notifiable Diseases Information System (Sinan) of the Rio de Janeiro State Health Department (SES/RJ) from 2011 to 2015 and from the INI Electronic Patient Record System (Sipec) from 2008 to 2015.

The study included the records of notifications comprising the sporotrichosis code of the International Statistical Classification of Diseases and Related Health Problems in its 10th version (ICD-10, codes B42 [and respective subcategories]) and cases of sporotrichosis confirmed by mycological examination at the INI. Data from the National Register of Health Establishments (CNES) were used to locate reporting units and population estimates from the Brazilian Institute of Geography and Statistics (IBGE) were used to calculate the incidence rate per inhabitant with the scale of 100 thousand.

### 2.2. Ethics

The study was approved by the Research Ethics Committee of INI/Fiocruz (CAAE 61114616.4.0000.5262), and the necessary measures were taken to avoid breach of patient confidentiality.

### 2.3. The State of Rio de Janeiro

The state of RJ is in southeastern Brazil, had a population of 15,989,929 inhabitants in the 2010 National Demographic Census, and occupies an area of 43,750,426 km² (https://cidades.ibge.gov.br/brasil/rj/panorama (accessed on 17 November 2021). The 92 municipalities of the state of RJ are divided into eight governmental regions: Metropolitana (Metropolitan region) with 22 municipalities, including the capital and Baixada Fluminense; Baixadas Litorâneas (Coastal Shore region); Norte Fluminense (Northern Fluminense region); Noroeste Fluminense (Northwest Fluminense region); Serrana (Mountain region), Centro-Sul Fluminense (Central South Fluminense region); Médio Paraíba (Middle Paraíba region); and Costa Verde (Green Coast region) (map of the state—Appendix A and socioeconomic characteristics—Appendix A).

### 2.4. Spatial Analysis

For the sake of our spatial analysis, after grouping all cases registered as sporotrichosis in the period from 2008 to 2015, at INI and notified in Sinan from 2011 to 2015, the duplications referring to INI patients notified in Sinan based on name, date of birth, mother’s name, and sex, were excluded. A deterministic procedure was firstly used (complete data equality), followed by probabilistic linkage using OpenReclink version 3.1 (http://reclink.sourceforge.net/ (accessed on 17 November 2021), then complemented by manual search for cases where probability was too low.

The addresses were converted into geographic coordinates using the ViconSaga platform, transformed into Shapefiles and inserted into the QGis and ArcGis^®^ programs, where the spatial analysis was performed. Kernel maps were prepared to identify areas with higher density of cases.

## 3. Results

From 2008 to 2015, 2992 patients were diagnosed with sporotrichosis at the INI. Of these, 2010 (67.2%) were female and 982 (32.8%) were male. The median age was 43 and the mode was 43 years (range 1 to 92 years). There were 1563 patients (52.2%) living in the city of Rio de Janeiro. Residents of municipalities in the Baixada Fluminense were 1162 (38.8%). Cases were registered in 43 (46.7%) of the 92 municipalities in RJ (map of the state—Appendix A).

In the SES/RJ notification system, 4412 cases of sporotrichosis were reported from 2011 to 2015, of which 1032 (23.4%) were INI patients. Of these, 2946 (66.7%) were female and 1465 (33.3%) were male. The median age was 44 and the mode was 47 years (range 1 month to 99 years). Cases were registered in 49 (53.2%) of the municipalities in RJ. Two patients lived outside the state of RJ, one in the state of Minas Gerais and the other in the state of Paraíba.

After exclusion of duplicates and the linkage of the databases (as described above), there were 6372 cases, of which 5369 (84.3%) addresses of the cases were geocoded between 2008 and 2015. Regarding the remaining 1003 (15.7%) cases, there was a loss due to incorrect filling or inconsistencies of the available databases. This loss could not be remediated by any algorithmic or non-algorithmic (case-by-case) procedure.

INI was the main notifying health establishment (1032; 23.4%), followed by the Duque de Caxias Municipal Health Center with 224 (5.1%) cases, Dom Walmor Polyclinic, Nova Iguaçu with 108 (2.4 %) cases and Vasco Barcelos Health Center, Nova Iguaçu with 78 (1.8%) cases. The density of cases in the metropolitan region was higher in areas close to notifying health units (Figure 1).

In the metropolitan region, over the period, expansion and emergence of new high-density areas were observed in the west and north regions of the capital and its outskirts. In 2008, high-density areas of cases were observed on the limits with the cities of Nova Iguaçu, Mesquita, Nilópolis, São João de Meriti and Duque de Caxias, in addition to the emergence of four other high-density areas in the west and north of the capital (Figure 2a). In 2011, as the notification system had been launched, an increase in the extension of areas with cases was observed throughout the metropolitan region, mainly in the west and north of the capital. As of 2012, several new high-density areas were observed in the west, north and center of the capital, in addition to municipalities of the metropolitan area (Duque de Caxias, São João de Meriti, Mesquita, Belford Roxo, Nova Iguaçu, Nilópolis and Queimados). In 2014, in addition to a drop in the total number of cases, there was a reduction in the number of high-density areas, with an area emerging in the south of the capital, corresponding to the Rocinha neighborhood. In 2015, the cases increased again, with the confluence of the previous medium- and high-density areas.

In addition to the metropolitan region, the formation of medium- and medium-high density areas was observed in the Baixada Litorânea and municipalities neighboring the metropolitan region. As of 2012, in São Gonçalo and Niterói, areas of medium density were formed and, in 2014, an area of high density was formed in Magé (Figure 2b).

The incidence calculated for the state of RJ, based on population estimates by the IBGE, was 4.4 per 100,000 inhabitants in 2011, 4.9 in 2012, 6.7 in 2013, 4.8 in 2014 and 6.2 in 2015. In the capital, the rate at the beginning of the study period was 6.3 and in the last year 6.1, reaching the peak in 2013 (8.3). Carmo, a municipality in the Mountain region, had the highest rate in the period, 55.7 in 2013.

Additional findings are presented as maps included in the Appendix A.

## 4. Discussion

Silva et al. analyzed 1848 cases, diagnosed with sporotrichosis from 1997 to 2007 [22]. In the present study, 6372 cases were identified in the state of Rio de Janeiro between 2008 and 2015. The numbers suggest a progression from 168 cases/year to 796.5 cases/year over the period under analysis, an increase of 374%. Such estimates should be viewed with caution, as the best estimates that could be obtained by now. They should be revised by future studies as additional databases with a more robust structure become available.

Until 2011, in both studies, cases from other health facilities besides the INI were not included. Furthermore, over the years, the population of RJ has increased and become more heterogeneous considering its geographic strata (https://cidades.ibge.gov.br/brasil/rj/panorama (accessed on 17 November 2021). The observed increase may have been confounded by demographic and/or geographic changes. However, even considering the abovementioned changes, the increase in cases has been pronounced.

The profile of patients, especially adult women, was similar to other studies regarding sporotrichosis by zoonotic transmission at RJ, possibly due to the greater exposure of women taking care of sick cats [2,22].

The spatial analysis allowed us to verify the spread of this mycosis to all regions of the state during the period under analysis as well as the expansion of the formerly called “sporotrichosis belt” beyond the limits between the capital and municipalities of the metropolitan area.

The capital is divided into south, north, west, and central zone. Among these zones, the south zone has the best housing and sanitation structure. The west zone stood out in our study for the large number of new high-density areas and persistent propagation, which can be explained in part by the urban sprawl (Appendix A) or even by the implementation of the family health strategy in the region [25], i.e., the actual increase in cases may have been partially confounded by a better reporting structure.

Secondary factors, such as disorderly growth, lack of adequate sanitary conditions and infrastructure, may be associated with the increase in cases. Alzuguir et al. described the cases at Duque de Caxias, municipality of the metropolitan area (from 2007 to 2016) where a high-density of the disease has been observed in areas with low per-capita income and deficient provision of basic sanitation services [26].

In our study, the socioeconomic characteristics were not studied locally, but it was possible to observe that the spread did not respect the boundaries, migrating to different areas, including some with recent urbanization within a rural environment. Furthermore, cats can walk for up to 6 km in the mating season [27] transmitting diseases to other neighborhood cats. In addition to the migration of humans carrying their pets (sometimes sick), this animal behavior could be responsible for the spread across the state.

The more consistent spread and increase in high-density areas starting in 2011 can be explained in part by the beginning of notification in the state [23,24]. Until then, the identified cases came exclusively from INI, with the majority of cases residing in cities in the metropolitan region. In addition, over the period there was a consistent increase in family health coverage in the city, from 486,000 inhabitants in 2008 to 2.7 million in 2015 [25]. However, even considering both confounders, the observed increase was too pronounced and many times greater than a stationary phenomenon observed from a different perspective, benefiting from a more comprehensive reporting system. Population coverage by health mobile teams (“agentes de saúde”) has expanded in the country, but not under such fast pace [25,28].

The 2014 drop does not seem consistent with the later expansion. Most probably, it is secondary to underreporting that year after the initial year of mandatory notification. This constitutes both a failure of the reporting system and a limitation of the present study that cannot be fully addressed. As in any study based on secondary data, reliable and valid data are the very basis of any analysis.

An area of greater density was observed in the southern zone in 2014, a region of the capital with few cases over a long period of time. Spatially, this area corresponds to the Rocinha neighborhood (a very dense low-income neighborhood, formerly defined as a “favela”). This relative densification seems to be an isolated event, but more detailed studies and proactive surveillance are needed in the region to determine the current situation of sporotrichosis in this location. As happens in many different infectious diseases, fully passive surveillance systems tend to underestimate infection rates and cases [29]. Under-reporting tends to be especially relevant among underserved, marginalized, and hard-to-reach populations [30].

The municipality of Carmo, despite having the highest incidence rate in the study, is a sparsely populated municipality (17,944 inhabitants in 2013). All 13 notifications occurred in 2013 and 2014 in the same neighborhood, possibly reflecting the occurrence of an outbreak from one or a few sources.

At the INI, 5113 cats were diagnosed from 1998 to 2018 [18]. However, estimates on cases among cats in RJ have been far from accurate. Cats are not systematically diagnosed by their owners (as suffering from a disease yet to be defined) and veterinary doctors. Even after a precise diagnosis, they might not be brought to the INI reference center. An unknown fraction of cats are stray cats. However, even when they are cared for by someone, people are usually too poor or not well informed about the disease and might simply not seek proper help. The figures from INI represent only a fraction of actual cases. This study was carried out using secondary databases, so it was not possible to assess whether the reported cases were related to zoonotic transmission. Recent publications in the literature bring evidence to believe that the vast majority of cases have occurred by zoonotic transmission, reported to occur in up to 90% of patients, with scratches or bites by infected cats in around 65% of them [2,10,11,12,18,22].

The municipality of Rio de Janeiro reported 12,751 confirmed cases of feline sporotrichosis from 2017 to 30 September 2021 [31] and the state of RJ reported 2616 confirmed human cases from 2019 to 2020 [32]. Although mandatory notification in Brazil is in place for several epizootic diseases, animal sporotrichosis remains outside this group. Recently, a resolution [33] ratified human sporotrichosis as a notifiable disease in RJ, including other mycoses in humans and animal sporotrichosis. The impact of such a resolution on the improvement in databases has yet to be analyzed.

There are two municipal free veterinary care units in the capital, which also accumulate a large number of cases, in addition to a helpline for the population to request inspections by veterinary professionals in places with suspected animals with the disease. Temporary government and non-governmental units operate intermittently and do not meet the increasing demand. In RJ, there is no standardization in the supply of human and animal treatment and there is frequently a shortage of medication. Many cats do not have guardians; others have volunteer caregivers in places called animal colonies.

The data presented in this study show the difficulty in controlling zoonotic sporotrichosis in RJ. Despite the public health measures adopted in the state, such as mandatory notification and decentralization of care, there are key obstacles in the effort to curb its spread. It is essential to improve surveillance of human cases, using comprehensive, proactive strategies. Notwithstanding, as there is no inter-human transmission of sporotrichosis, it is essential to intensify the control of the animal disease, not only in RJ, but also its surveillance throughout the country.

## Figures and Tables

**Figure 1 jof-08-00434-f001:**
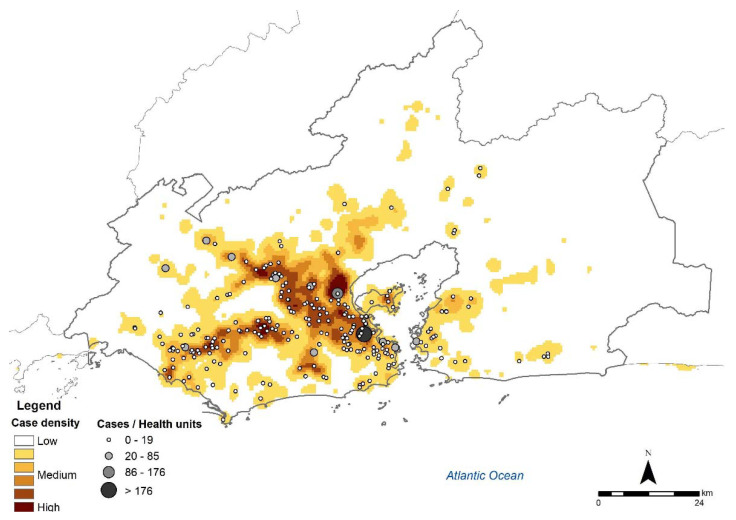
Cases of human sporotrichosis from 2008 to 2015 and notifying health units in the metropolitan region of the state of Rio de Janeiro (Source: SES-RJ, Sipec, CNES. Limits of neighborhoods: Demographic Census 2010, IBGE. Map created by the authors).

**Figure 2 jof-08-00434-f002:**
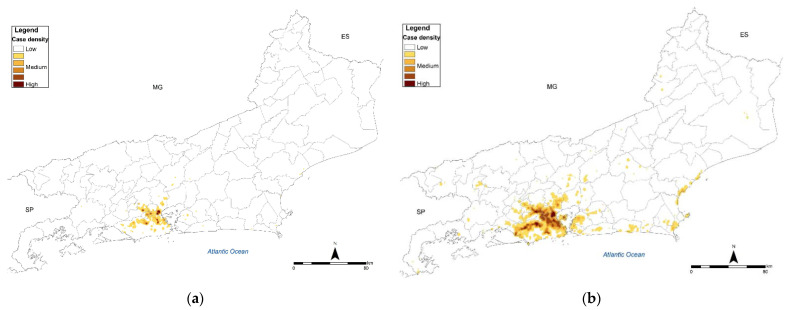
Spatial distribution (Kernel map) of sporotrichosis cases in the state of Rio de Janeiro (**a**) 2008; (**b**) accumulated from 2008 to 2015 (Source: SES-RJ, Sipec. Limits of neighborhoods: Demographic Census 2010, IBGE. Map created by the authors).

## Data Availability

Not applicable.

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
