# Peer review of "A Spatial Analysis of the Spread of Hyperendemic Sporotrichosis in the State of Rio de Janeiro, Brazil"

_jof, 2022, doi:10.3390/jof8050434_

Round 1
Reviewer 1 Report
- The author reported an increase number of sporotrichosis which spread in all regions of the state. Nevertheless, there was no clarify associated factors that made the case increasing such as occupation, environmental change, geographic areas….. etc.
- In non-Brazilian, it is difficult to understand the different between each region that increase the cases of sporotrichosis. Is there any things difference (people, geographic, occupation,….etc)
- In this study mentioned that increase cat sporotrichosis was one of the factors that increase human sporotrochosis. Although I did not find the correlation about cat and sporotrichosis patients. There was no data that show in this study that the patients were the owner of cat sporotrichosis.
- In my opinion, cat is not the only one factors of sporotrichosis. Sporotrichosis can be found in environment (such as soil, plants). So control of the animal disease alone may be not reduce the diseases.
Author Response
We thank you for the great considerations to our study. Please, find here our answers to the questions on the manuscript jof-1682658 - A spatial analysis of the spread of hyperendemic sporotrichosis in the state of Rio de Janeiro, Brazil. We managed to answer and solve all the possible issues.
- The author reported an increase number of sporotrichosis which spread in all regions of the state. Nevertheless, there was no clarify associated factors that made the case increasing such as occupation, environmental change, geographic areas….. etc.
Answer: Thank you for pointing out this issue. The state of Rio de Janeiro is not so vast, but its regions differ a lot in geographic, socioeconomic, and demographic aspects. Please, see if lines 203-208 help explain the mechanisms believed to be part of this spread.
- In non-Brazilian, it is difficult to understand the different between each region that increase the cases of sporotrichosis. Is there any things difference (people, geographic, occupation,….etc)
Answer: We added a supplementary table summarizing the aspects of each region (cited in line 102). We hope it clarifies in this understanding. Also, we made other adjustments in lines 169-170 and 192-193.
- In this study mentioned that increase cat sporotrichosis was one of the factors that increase human sporotrochosis. Although I did not find the correlation about cat and sporotrichosis patients. There was no data that show in this study that the patients were the owner of cat sporotrichosis.
Answer: Some individual information could not be obtained from the records. We added a couple of sentences to address this limitation of the study. Please, see lines 243-248.
- In my opinion, cat is not the only one factors of sporotrichosis. Sporotrichosis can be found in environment (such as soil, plants). So control of the animal disease alone may be not reduce the diseases.
Answer: In Brazil, starting in Rio de Janeiro in the end of the 90s, the zoonotic pathway became the main mode of transmission. Environmental sources represent just a small part. We do agree that controlling the disease in the cats does not eliminate de disease in humans, but it would considerably decrease it. We aimed to address this issue bit more in lines 203-208 and 245-248.
Reviewer 2 Report
REVIEW REPORT
ABSTRACT
The sentence “The largest outbreak has been 17 described in South Africa, affecting 3,000 workers.” Is not related to this work and seems unnecessary. Can be omitted from abstract since it would be more relevant in the ‘background’ section of the main manuscript.
INTRODUCTION
I am not sure if we can use abbreviation for proper place names – if the journal allows it then it should be alright – example: Rio de Janeiro (RJ)
In line 51-52: Please explain what are the “domestic duties” that women do to expose them to this fungi? “Mostly middle-aged women involved in domestic duties have been infected,….”
RESULTS
Which age group is the mode? It would be interesting to see which age group (perhaps grouped every 10 years), has the highest number of patients?
Are the cases calculated and presented in the results involve individual patients? Could there have been repeat infections in the same individual patient over the years? How many are locals and foreigners?
DISCUSSION
Referring to lines 183 & 184, the authors mentioned “The profile of patients, especially adult women, was similar to other studies regarding sporotrichosis by zoonotic transmission at RJ, possibly due to the greater exposure of women taking care of sick cats”: Is this just a speculation? Did the authors consider looking at the profiles of the individuals infected with sporotrichosis in this study? Relation to types of profession to analyse the risk factors objectively?
What are the proportion of these cases in RJ is caused by direct contact with cats and what are the other types of exposure?
The spatial study has been thoroughly done and well written.
Author Response
We thank you for the great considerations to our study. Please, find here our answers to the questions on the manuscript. We managed to answer and solve all the possible issues.
REVIEW REPORT
ABSTRACT
The sentence “The largest outbreak has been 17 described in South Africa, affecting 3,000 workers.” Is not related to this work and seems unnecessary. Can be omitted from abstract since it would be more relevant in the ‘background’ section of the main manuscript.
Answer: Thank you. We have deleted the sentence.
INTRODUCTION
I am not sure if we can use abbreviation for proper place names – if the journal allows it then it should be alright – example: Rio de Janeiro (RJ)
Answer: Thank you. We chose to keep it because we did not find specific restrictions in the journal's rules. Of course, we can change it to full form if required.
In line 51-52: Please explain what are the “domestic duties” that women do to expose them to this fungi? “Mostly middle-aged women involved in domestic duties have been infected,….”
Answer: In fact, there are no specific duties in the process of infection, but the cohabitation with infected cats. To clarify this understanding, we changed the sentence to: “Mostly middle-aged women involved in domestic exposure to cats while taking care of the house have been infected,…”. Please, see lines 49-52.
RESULTS
Which age group is the mode? It would be interesting to see which age group (perhaps grouped every 10 years), has the highest number of patients?
Answer: We calculated the mode of both samples and added the data in the manuscript. Please, see lines 118 and 124.
Are the cases calculated and presented in the results involve individual patients?
Answer: Some individual information could not be obtained from the records (secondary database) but there are individual records with some variables for each case, like age, sex, notification heath service, and place of residence.
Could there have been repeat infections in the same individual patient over the years?
Answer: No patients’ ID were repeated in our database, except those filtered and excluded when both databases were merged for the analyses. They represented a duplication of data, not reinfection. Thus, although not with absolute certainty, it is probable that there were no cases of reinfection or duplicate notifications in this study.
How many are locals and foreigners?
Answer: Two patients lived out of the state of Rio de Janeiro. We added this information in lines 125-127.
DISCUSSION
Referring to lines 183 & 184, the authors mentioned “The profile of patients, especially adult women, was similar to other studies regarding sporotrichosis by zoonotic transmission at RJ, possibly due to the greater exposure of women taking care of sick cats”: Is this just a speculation? Did the authors consider looking at the profiles of the individuals infected with sporotrichosis in this study? Relation to types of profession to analyse the risk factors objectively?
Answer: Some individual information could not be obtained from the records. We added a couple of sentences to address this limitation of the study. Please, see lines 243-248.
What are the proportion of these cases in RJ is caused by direct contact with cats and what are the other types of exposure?
Answer: This proportion ranges from 80 to 90% in publications in the scientific literature. In some cases, there is no obvious trauma, but in around 65% there is a trauma mentioned by the patients. We hope the text added in lines 243-248 helps.
The spatial study has been thoroughly done and well written.
Answer: Thank you.